# Phasic locus coeruleus activity enhances trace fear conditioning by increasing dopamine release in the hippocampus

**Jacob H Wilmot[1,2], Cassiano RAF Diniz[2], Ana P Crestani[2], Kyle R Puhger[1,2], Jacob Roshgadol[2,3], Lin Tian[4], Brian Joseph Wiltgen[1,2]\***

[1]Department of Psychology, University of California, Davis, Davis, United States; [2]Center for Neuroscience, University of California, Davis, Davis, United States; [3]Department of Biomedical Engineering, University of California, Davis, Davis, United States; [4]Department of Biochemistry and Molecular Medicine, University of California, Davis, Davis, United States

**\*For correspondence:**
bjwiltgen@ucdavis.edu

**Competing interest:** The authors declare that no competing interests exist.

**Abstract** Locus coeruleus (LC) projections to the hippocampus play a critical role in learning and memory. However, the precise timing of LC-hippocampus communication during learning and which LC-derived neurotransmitters are important for memory formation in the hippocampus are currently unknown. Although the LC is typically thought to modulate neural activity via the release of norepinephrine, several recent studies have suggested that it may also release dopamine into the hippocampus and other cortical regions. In some cases, it appears that dopamine release from LC into the hippocampus may be more important for memory than norepinephrine. Here, we extend these data by characterizing the phasic responses of the LC and its projections to the dorsal hippocampus during trace fear conditioning in mice. We find that the LC and its projections to the hippocampus respond to task-relevant stimuli and that amplifying these responses with optogenetic stimulation can enhance long-term memory formation. We also demonstrate that LC activity increases both norepinephrine and dopamine content in the dorsal hippocampus and that the timing of hippocampal dopamine release during trace fear conditioning is similar to the timing of LC activity. Finally, we show that hippocampal dopamine is important for trace fear memory formation, while norepinephrine is not.

## eLife assessment

This is an **important** study examining the neural profile of weak and strong fear memories using a variety of imagining and interrogation neural techniques. The data are **convincing** in detailing the neural profile of neutral, aversive and fear conditioned stimuli in the LC and its input to the dorsal hippocampus and support the conclusion that dopaminergic input from the LC is the key instigator of trace fear conditioning in hippocampus. This paper is of interest to behavioural and neuroscience researchers studying learning, memory and neural networks.

## Introduction

The locus coeruleus (LC) supports an array of cognitive processes by modulating brain-wide arousal states and responding to salient events in the environment. LC neurons accomplish this via two distinct modes of activity: tonic and phasic. Changes in the frequency of tonic activity are associated with corresponding changes in psychological state. Low tonic activity (~0–2 Hz) is associated with drowsiness or sleep and increasing levels of tonic activity (~3–10 Hz) are associated with increased arousal,

**eLife digest** Our brains are more likely to remember activities or incidents that stand out from typical day-to-day experiences. For instance, if your phone is stolen on the way to work, you will have a stronger memory of this experience compared to other uneventful commutes. These are known as salient events and can be emotional, surprising, or even just out of the ordinary.

During salient events, an area of the brain known as the hippocampus receives chemicals called neuromodulators from other parts of the brain. These neuromodulators enhance the formation of the memory by modifying how neurons connect together in the hippocampus.

One of the regions that signals to the hippocampus – called the locus coeruleus – was thought to enhance memory by releasing the neuromodulator norepinephrine. Recent studies indicate that the locus coeruleus also releases a second neuromodulator called dopamine. However, it remained unclear what causes the locus coeruleus to release dopamine, and what effect this neuromodulator has on the hippocampus.

To investigate these questions, Wilmot et al. recorded and manipulated the activity of the locus coeruleus in the brains of mice experiencing salient, fearful events. The mice were exposed to a sound and, a few seconds later, a shock to the foot to illicit the formation of an aversive salient memory. If the next day, the mice responded to just the sound as if they were expecting a shock, this indicated they had remembered the aversive experience.

Wilmot et al. observed that neurons in the locus coeruleus were active during the salient event, resulting in increased dopamine in the hippocampus. When the activity of these neurons was forcefully increased during relatively non-salient events, such as a quiet tone and a very mild shock, the animals still showed strong memory formation. Finally, blocking the action of dopamine in the hippocampus substantially affected memory formation, whereas blocking the action of norepinephrine did not have the same effect.

These findings suggest that the locus coeruleus enhances the memory of salient events by increasing the levels of dopamine in the hippocampus not norepinephrine, as was previously thought. Developing a better understanding of how the locus coeruleus regulates memory may lead to improved treatments for various neurological disorders, like Alzheimer's disease, which are associated with neuromodulators taking on different roles in the hippocampus.

progressing from exploration and task engagement to agitation and anxiety states (*Aston-Jones and Cohen, 2005*). Bouts of phasic activity (~10–20 Hz) are most often observed during intermediate levels of tonic activity in response to salient environmental events (*Aston-Jones et al., 1999*). When locked to task-relevant events, these phasic responses are associated with improved cognitive performance across a variety of tasks (*Aston-Jones and Cohen, 2005* for review).

A large body of research supports the idea that the LC is critically involved in memory formation. Monoamine depletion in the LC, noradrenergic and dopaminergic antagonism in multiple brain regions, and direct inhibition of the LC all impair memory across a number of tasks (*Giustino and Maren, 2018*; *Lisman and Grace, 2005*; *Selden et al., 1990*; *Uematsu et al., 2017*; *Wagatsuma et al., 2018*). Conversely, LC stimulation as well as dopamine and norepinephrine agonism can enhance memory (*Bach et al., 1999*; *Kempadoo et al., 2016*; *Packard and White, 1991*; *Packard and White, 1989*; *Sara and Devauges, 1988*). Although the LC is typically thought to modulate neural activity via the release of norepinephrine, several recent studies have suggested that it may also release dopamine into the hippocampus (HPC) and other cortical regions (*Devoto and Flore, 2006*; *Smith and Greene, 2012*). In some cases, it appears that dopamine release from LC into the HPC may be more important for memory than norepinephrine (*Kempadoo et al., 2016*; *Takeuchi et al., 2016*; *Wagatsuma et al., 2018*).

However, studies examining the effect of LC on HPC-dependent memory have generally not distinguished between tonic and phasic LC activity or between dopamine and norepinephrine signaling. In some cases, this distinction is prevented by the use of temporally imprecise manipulations like lesions and drug infusions. In others, it is difficult to determine the importance of time-locked phasic firing because of the use of spatial memory tasks that are commonly used to assess hippocampus (HPC)-dependent memory. Because space is a temporally diffuse stimulus, it is difficult to know the exact

moments during spatial learning when phasic responses might be important. This does not rule out the possibility that phasic LC activity and dopamine signaling is important for spatial learning, but the lack of experimental control over the animals' sampling of the relevant stimuli makes this possibility opaque to examination. Studies that have used discrete reward learning tasks to examine dopamine function in the HPC have also not typically distinguished between tonic and phasic release.

To address these issues, we used a trace fear conditioning task in which animals learn an association between a neutral and aversive stimulus that are separated in time. Because trace fear conditioning (a) requires intact hippocampal activity (*Chowdhury et al., 2005*; *Raybuck and Lattal, 2014*; *Wilmot et al., 2019*) and (b) involves learning about discrete, well-controlled stimuli, it is ideally suited to characterize the rapid temporal dynamics of LC activity and LC-HPC communication during learning and examine their role in long term memory formation. Here, we used fiber photometry and optogenetics to make temporally precise observations and manipulations of LC activity during learning in a trace fear conditioning task. We extend this data by providing direct evidence that LC activity increases both dopamine and norepinephrine content in the HPC and by showing that dopamine release is important for trace fear memory formation, but the release of norepinephrine is not.

## Results

### The locus coeruleus responds to neutral and aversive environmental stimuli

Before examining LC activity during learning, we sought to determine whether the LC exhibits phasic responses to the basic types of stimuli used in trace fear conditioning. Because previous research indicates LC activity may be modulated by the salience, or intensity, of environmental stimuli (*Aston-Jones and Bloom, 1981*; *Aston-Jones and Cohen, 2005*; *Vazey et al., 2018*), we began by examining the responses of the locus coeruleus to neutral auditory stimuli of varying intensities (55 dB-95dB).

To observe LC responses to auditory stimuli, we expressed the genetically encoded calcium indicator GCaMP6s specifically in the LC of TH-Cre transgenic mice and implanted an optical fiber above the injection site to allow fiber photometric recordings of bulk LC activity (*Figure 1A* and *Figure 1B*). After recovery from surgery, GCaMP fluorescence in LC was measured as the mice were exposed to five interleaved presentations of a 3000 Hz pure tone at each dB level (25 total presentations). Small phasic responses were seen at each dB level, but responses increased with tone intensity, with responses to the 95 dB tone most pronounced (*Figure 1C and D*), confirming that the magnitude of the LC response to an auditory stimulus is modulated by its intensity.

We next examined LC responses to aversive stimuli. To determine whether LC responses are modulated by the intensity of an aversive stimulus, we recorded calcium activity in the LC while mice received foot shocks of varying intensity (0–0.8mA). Consistent with the idea that LC activity is modulated by the salience of environmental stimuli, we found that LC responses to foot shock were much larger than the previously observed responses to neutral auditory stimuli (*Figure 1C–F*). Additionally, consistent with previous reports, the size of the LC response to footshock increased with foot shock intensity (*Figure 1E and F*; *Chen and Sara, 2007*; *Hirata and Aston-Jones, 1994*; *Rasmussen and Jacobs, 1986*; *Uematsu et al., 2017*). The results of these initial experiments demonstrate that the LC responds to both neutral and aversive stimuli and that LC response magnitude is positively correlated with the salience and/or valence of the stimulus.

### Locus coeruleus activity is modulated by learning

Having demonstrated that the locus coeruleus is responsive to salient environmental events, we next sought to determine whether its responses are modulated by learning. Again, we began by using fiber photometry to monitor LC responses to a neutral auditory stimulus. If the LC responds to novelty or salience, we would expect the responses to diminish over repeated experiences with a stimulus as the animal habituates to its presence (*Thompson and Spencer, 1966*). To test this idea, we recorded activity in the LC over the course of 3 days as animals were repeatedly exposed to the same tone (85 dB, 3 KHz; *Figure 2A*). As expected, LC responses to the tone were largest on the first day and gradually reduced in magnitude over the next 2 days (*Figure 2B and C*). The LC also responded to tone termination, and while this response also appeared to habituate across days, the effect did not reach significance (Main effect of day $F(2,8) = 3.36$, $p=0.087$). This finding is consistent with prior work

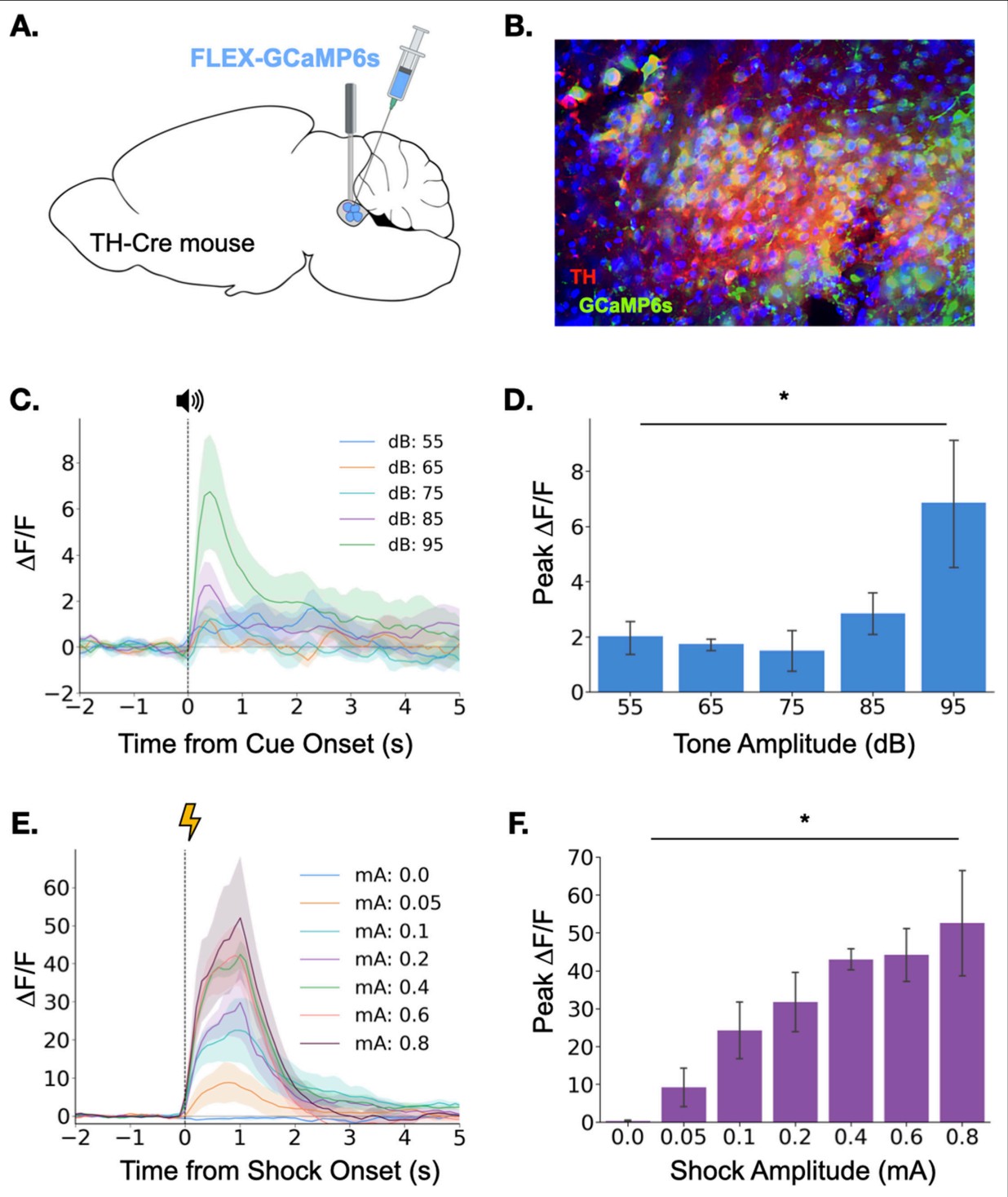

**Figure 1.** Locus coeruleus responses to neutral and aversive stimuli. (**A**) Schematic of virus infusion and fiber implant in LC. FLEX-GCaMP6s was infused into the LC of TH-Cre mice (n=4) and an optical fiber was implanted just above the injection site. (**B**) GCaMP6s expression in a sagittal section of LC. Green = GCaMP, Red = Tyrosine Hydroxylase. (**C**) Fiber photometry traces of LC responses to tone onset at varying dB levels. Dashed line indicates tone onset. (**D**) Peak ΔF/F during tone onset differed across tone amplitudes ($F_{(4,12)} = 3.36$, $p<0.05$). Mean +/- SEM. (**E**) Fiber photometry traces of LC responses to shock onset at varying mA levels. Dashed line indicates shock onset. (**F**) Peak ΔF/F at shock onset increased across shock amplitudes ($F_{(6,18)} = 6.46$, $p<0.001$). Mean +/- SEM.

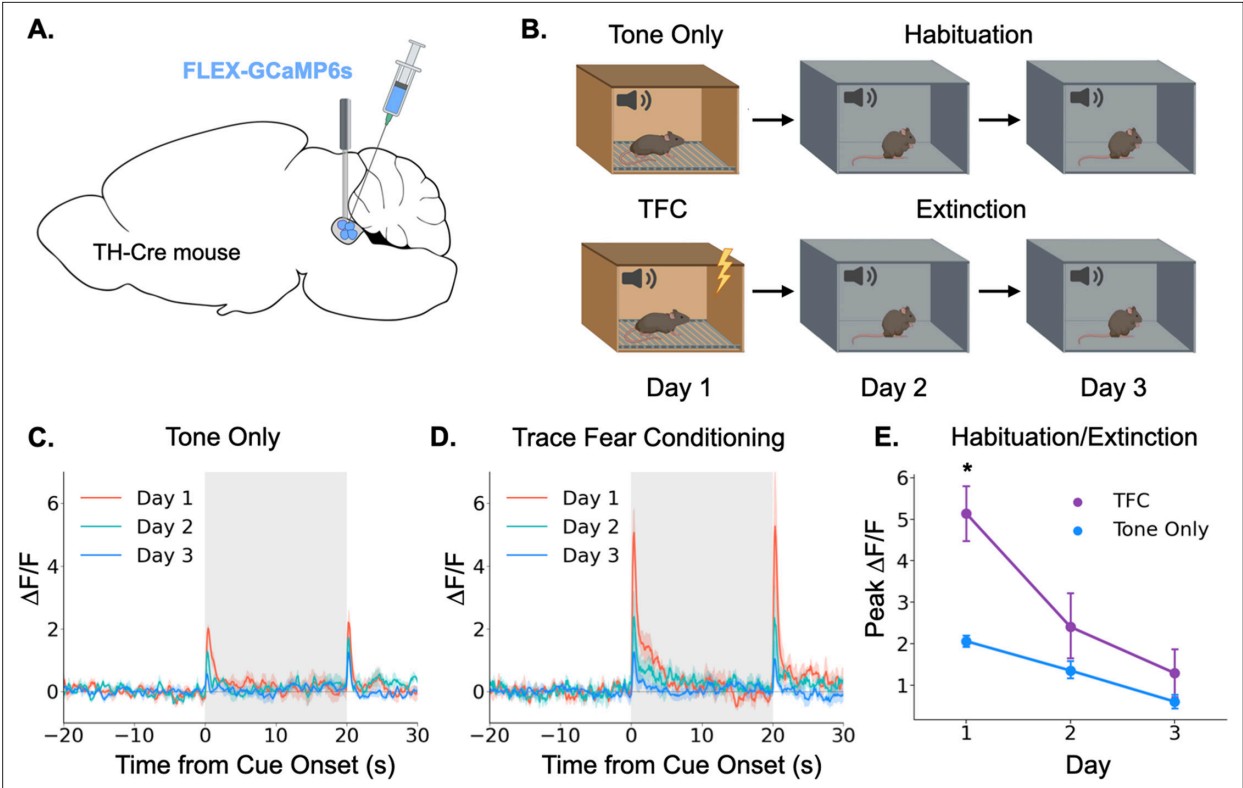

**Figure 2.** Locus coeruleus responses are modified by learning. (**A**) Cre-dependent GCaMP6s was infused into the LC of Th-Cre mice (TFC, n=4; tone only, n=5) and a fiber optic was implanted above the injection site. (**B**) Schematic of behavioral conditions. On Day 1, mice received either 10 tone presentations or 10 tone-shock pairings (trace fear conditioning) in Context A. On Day 2, all mice received 10 tone presentations in Context B. On Day 3, all mice received 20 tone presentations in Context B. (**C**) Fiber photometry traces of LC responses to tone onset and termination in tone only animals across the 3 experimental days. The response to tone onset decreased significantly across days ($F_{(2,8)}$ = 18.3, p<0.01). The response to tone termination did not change across days ($F_{(2,8)}$ = 3.36, p>0.05).(**D**) Fiber photometry traces of LC responses to tone onset and termination in trace fear conditioned animals across the 3 experimental days. The responses to both tone onset and tone termination decreased significantly across days (Onset: $F_{(2,6)}$ = 13.8, p<0.01; Termination: $F_{(2,6)}$ = 9.84, p<0.05). (**E**) Peak ΔF/F during tone onset for trace fear conditioned and tone only mice across the 3 experimental days differed significantly on day 1, but not days 2 or 3 (Day x Group interaction: $F_{(2,14)}$ = 6.45, p<0.05; Day 1, TFC vs tone only: $t_{(7)}$ = 3.74, p<0.05; Day 2: $t_{(7)}$ = 1.24, p>0.05; Day 3: $t_{(7)}$ = 1.09, p>0.05).

showing that animals process the termination of an auditory CS as a distinct, salient event that itself can undergo conditioning (*Sommer-Smith, 1967*; *Sommer-Smith et al., 1962*).

Next, we examined the effect of associative fear learning on LC responses in a second group of mice that underwent the same procedures, except that on the first day they received ten trace fear conditioning trials instead of unpaired tone presentations (*Figure 2D*). On the second and third days, mice underwent extinction trials in which no shocks were administered. Trace conditioning trials consisted of a 20 s tone followed by a 20 s stimulus free trace interval ending in a 2 s 0.2mA foot-shock. The LC responses in these animals followed a similar pattern to that observed in the tone only group, with activity on Day 1 being the largest and gradually decreasing across days as the mice extinguished. However, the magnitude of the LC responses was much larger in mice that underwent trace fear conditioning on Day 1 compared to tone alone presentations (*Figure 2E*). We also found that the response to tone termination significantly decreased across the extinction sessions. These data suggest that LC responses can be modified by associative learning – LC responses to an inherently neutral stimulus are larger when the stimulus is paired with an aversive outcome than when the stimulus is experienced in isolation. Additionally, as this association is extinguished and the tone becomes less predictive of an aversive outcome, the LC response is reduced – there was no significant difference between fear conditioned and tone only mice on the second or third days of behavior when

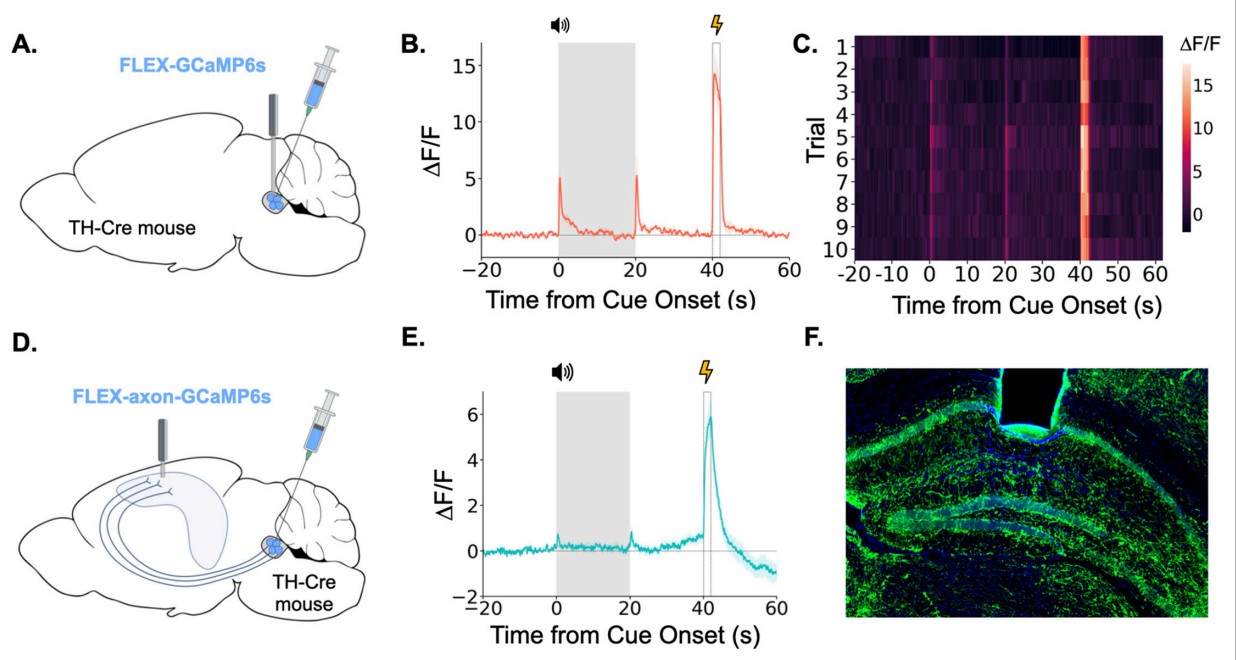

**Figure 3.** LC neurons and LC-dHPC projections respond to trace conditioning. (**A**) Cre-dependent GCaMP6s was infused into the LC of TH-Cre mice (n=4) and an optical fiber implanted above the infusion site. (**B**) Fiber photometry trace showing LC $Ca^{2+}$ activity during trace fear conditioning. $Ca^{2+}$ activity increased significantly at tone onset (t(3) = 6.30, p<0.01) and shock (t(3) = 9.82, p<0.01). Activity increased numerically at tone termination, but did not reach significance (t(3) = 2.91, p=0.062) However, as seen in *Figure 2*, this response did significantly decrease with extinction. (**C**) Heatmap of trial by trial $Ca^{2+}$ activity in LC across 10 trace fear conditioning trials. Activity was consistent across trials. (**D**) Cre-dependent axon-GCaMP6s was infused into the LC of TH-Cre mice (n=11) and an optical fiber was implanted in the dHPC. (**E**) Fiber photometry traces showing axon-GCaMP responses to trace conditioning. $Ca^{2+}$ activity significantly increased at tone onset tone onset (t(10) = 2.77, p<0.05), tone termination (t(10) = 3.03, p<0.05), and shock (t(10) = 8.09, p<0.01). (**F**) Image showing axon-GCaMP expression (green) in dHPC. Blue = DAPI.

neither group was receiving footshocks. These results are consistent with previous recording studies that found individual LC neurons can acquire a learned response to the conditional stimulus during training that disappears during extinction (*Rasmussen and Jacobs, 1986*; *Sara and Segal, 1991*).

## Locus coeruleus terminals in dorsal hippocampus exhibit phasic responses during trace fear conditioning

Taken together, the above data show that the LC exhibits large bursts of activity during trace fear conditioning at the tone onset, tone termination, and during the shock. LC responses at these key task-related events are overtly evident in LC GCaMP traces averaged across trials and on each trial individually (*Figure 3A, B and C*). We next examined whether the specific projection from the LC to the dorsal hippocampus (dHPC) exhibits similar phasic responses during trace fear conditioning. To do this, we infused a cre-dependent AAV encoding an axon-enriched calcium sensor, axon-GCaMP6s, into the LC of TH-Cre mice and implanted an optical fiber just above dorsal CA1 (*Figure 3D and F*; *Broussard et al., 2018*).

When we recorded activity from LC-HPC projections during trace fear conditioning, we found that these projections responded to all relevant trace conditioning events: tone onset, tone termination, and shock (*Figure 3E*). These results confirm that the phasic responses observed in LC during trace conditioning are also present in the specific subset of LC axons that project directly to the dHPC and suggest that LC-dHPC projections may be important for memory formation.

## Phasic locus coeruleus activation enhances trace fear conditioning

Our fiber photometry experiments provide strong correlational evidence that phasic LC responses are involved in trace fear learning via the release of catecholamines, but whether this response enhances

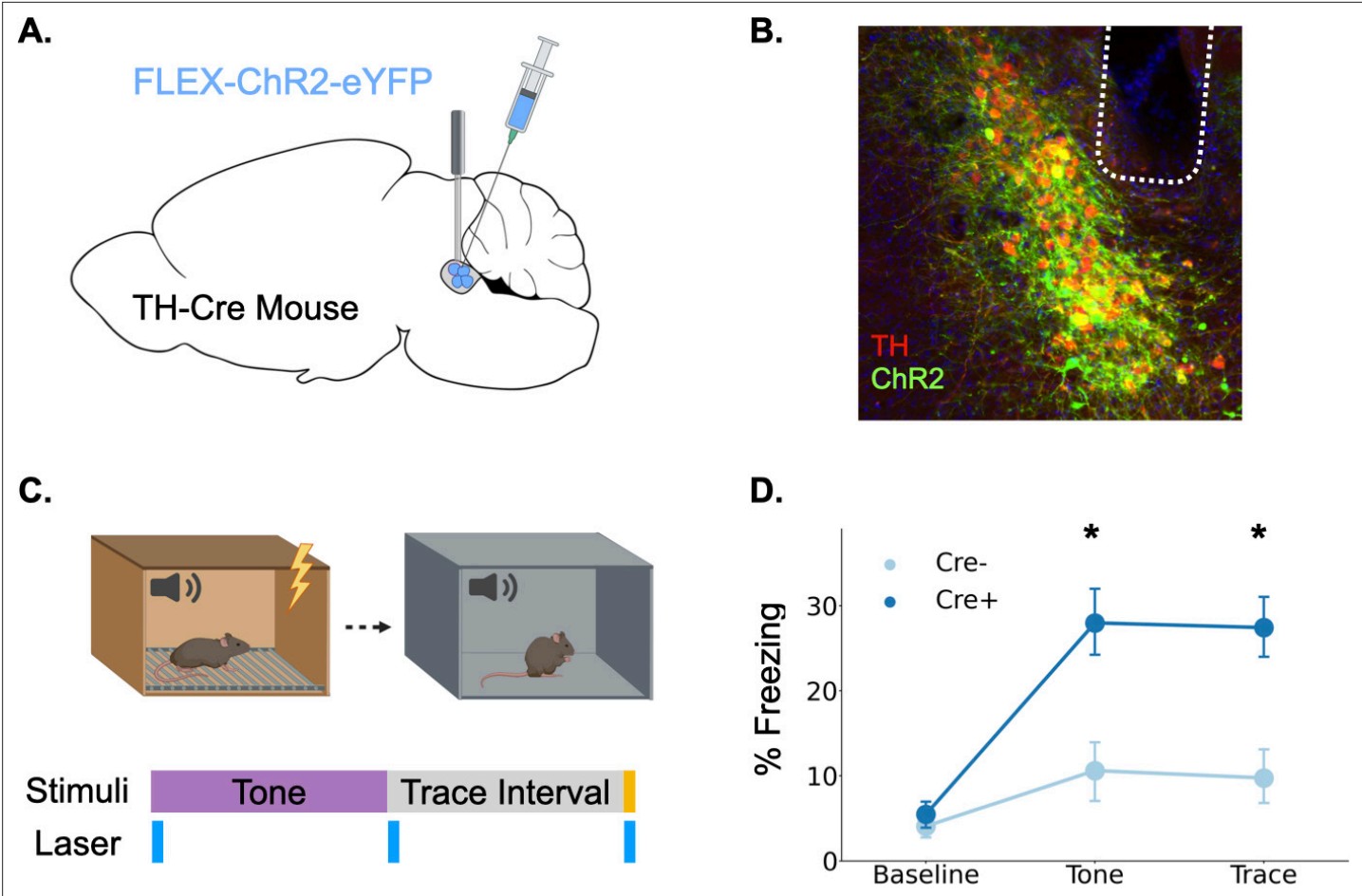

**Figure 4.** Phasic activation of the locus coeruleus enhances long-term memory formation. (**A**) Cre-dependent ChR2 was infused into the LC of TH-Cre+ (n=7) and TH-Cre- (n=7) mice and optical fibers were implanted above the infusion site. (**B**) Expression of ChR2 in LC. Green = ChR2 eYFP, Red = tyrosine hydroxylase, Blue = DAPI. (**C**) Schematic of behavioral procedures. Animals were trained in trace fear conditioning in Context A on Day 1 and tested in Context B on Day 2. During training, 20 Hz blue light was delivered to the locus coeruleus for two seconds at the beginning and end of the tone as well as during the shock. (**D**) Memory performance for the two groups during the tone test as indexed by freezing behavior. The groups did not differ at baseline, but the Cre +mice expressing ChR2 froze significantly more during the tone and the post-tone period corresponding to the trace interval on the training day (significant Group x Trial Phase interaction: $F_{(2,24)}$ = 5.28, $p<0.05$; tones ($t_{(12)}$ = 3.08, $p<0.05$);traces ($t_{(12)}$ = 3.52, $p<0.01$)).

learning, as is often proposed, remains unknown (*Giustino and Maren, 2018*; *Likhtik and Johansen, 2019*; *McGaugh, 2004*; *Sears et al., 2013*; *Takeuchi et al., 2016*). To determine whether phasic activation of LC enhances trace fear conditioning, we infused a cre-dependent version of the excitatory opsin ChR2 into the LC of TH-Cre mice or their wild-type littermates and optical fibers were implanted above the infusion site (*Figure 4A and B*). After recovery from surgery, the mice underwent trace fear conditioning using a protocol with low tone (65 dB) and shock intensity (0.2mA) to produce weak learning that would allow us to uncover an enhancement with LC stimulation, as indicated by the fact that these stimulus intensities produced relatively weak LC responses (*Figure 1*). During the training session, 20 Hz blue light was delivered to the locus coeruleus in three two-second periods beginning at each of the trial events when consistent LC responses were observed in fiber photometry experiments: tone onset, tone termination, and shock onset (*Figure 4C*).

The next day, animals underwent a memory test in a novel context in which they were repeatedly exposed to the training tone (*Figure 4D*). Freezing during the tone and the post-tone period was used as an index of fear memory retrieval. Phasic LC stimulation at the onset of all learning-related events significantly enhanced long-term memory formation. Cre-positive mice expressing ChR2 froze

significantly more in the memory test during both the tones and the 20 s post-tone period corresponding to the trace-interval on the training day. These data indicate that increased phasic activation of the LC at specific learning-related time points during trace conditioning is sufficient to enhance long-term memory formation.

## The locus coeruleus drives both dopamine and norepinephrine release in the dorsal hippocampus

Our data suggest that LC projections to the hippocampus are activated by salient learning-related events and that this LC activation can enhance long term memory formation. However, it remains unclear which neurotransmitter the LC releases into the dHPC during these responses. Canonically, the LC is known as the primary source of norepinephrine in the forebrain (*Jones and Moore, 1977*; *Lindvall and Björklund, 1974*; *Pickel et al., 1974*; *Schwarz and Luo, 2015*). However, recent evidence suggests that the LC can co-release norepinephrine and dopamine into the hippocampus and other cortical areas (*Devoto and Flore, 2006*; *Kempadoo et al., 2016*; *Smith and Greene, 2012*; *Takeuchi et al., 2016*; *Wagatsuma et al., 2018*).

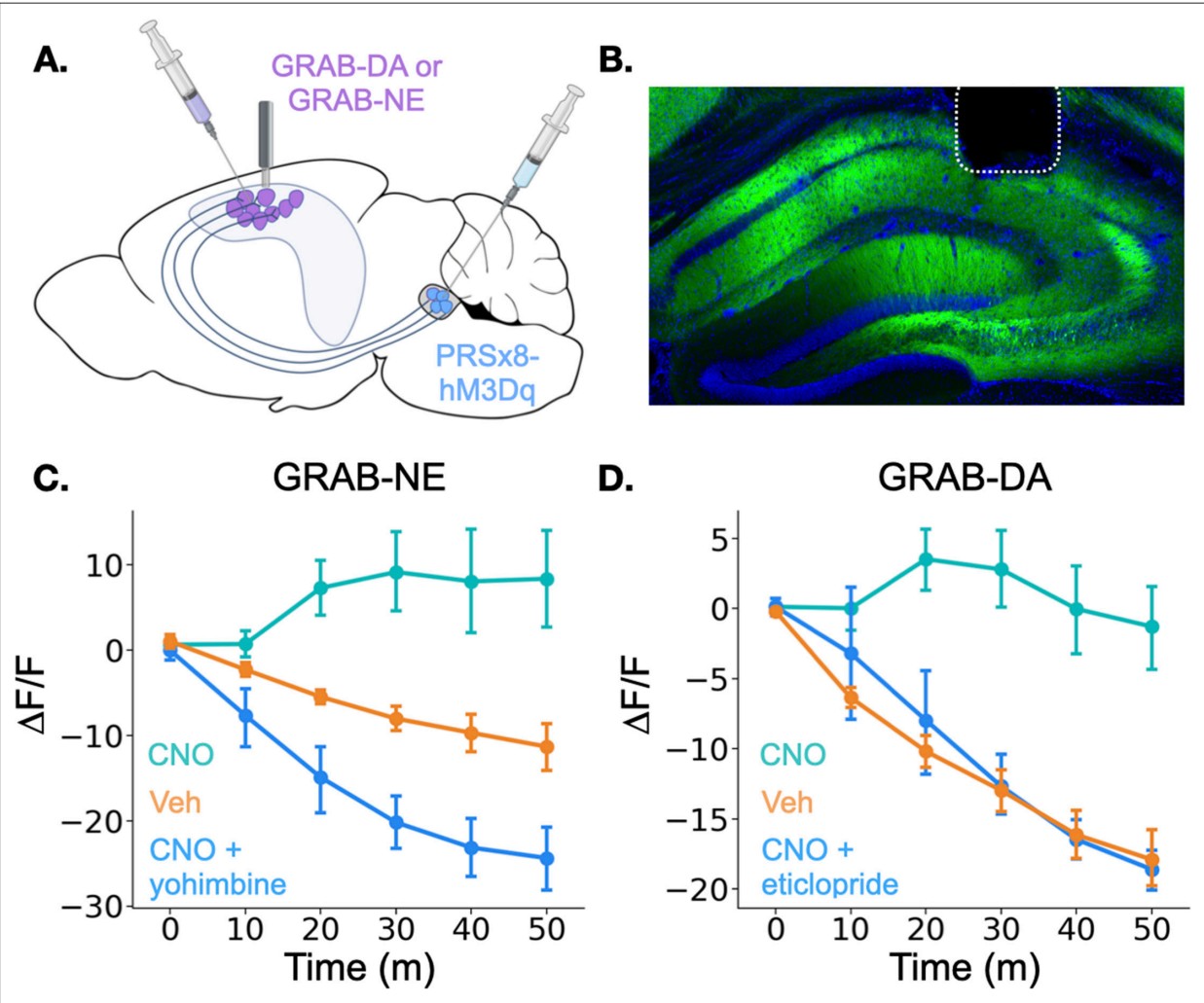

**Figure 5.** Locus coeruleus activation increases norepinephrine and dopamine in the dorsal hippocampus. (**A**) GRAB-DA (n=6) or GRAB-NE (n=5) was infused into the dorsal hippocampus and an optical fiber was implanted in CA1. PRSx8-hM3Dq was infused into the locus coeruleus. (**B**) GRAB-NE expression and fiber placement in dHPC. (**C**) CNO injections increased dHPC norepinephrine relative to vehicle or combined CNO and yohimbine injections (Treatment x Time interaction: $F_{(10,40)} = 12.5$, $p<0.05$). (**D**) CNO injections increased dHPC dopamine relative to vehicle or combined CNO and eticlopride injections (Treatment x Time interaction: $F_{(10,50)} = 6.45$, $p<0.05$).

To test this possibility, we expressed the genetically encoded dopamine and norepinephrine sensors GRAB-DA and GRAB-NE, respectively, in the dorsal CA1 of separate groups of mice and implanted optical fibers above the injection site (*Figure 5A*; *Feng et al., 2019*; *Sun et al., 2020*). In the same animals, we expressed the excitatory DREADD hM3Dq in LC under the control of the PRSx8 promoter, which can be used to drive selective expression in LC neurons (*Abbott et al., 2009*; *Hwang et al., 2001*; *Vazey and Aston-Jones, 2014*). After the animals recovered from surgery, we recorded fluorescence as animals freely explored a conditioning chamber. After 10 min of baseline recording, we injected mice with either CNO, to activate LC neurons, or vehicle and continued to record fluorescence for 50 more minutes. Mice that received CNO the first day were treated with vehicle on the second day and vice versa. On a third day, to rule out non-specific effects of CNO and confirm the function of the sensors, animals received injections of both CNO and an antagonist for the receptor that serves as the backbone of the sensors. GRAB-NE animals were injected with the α–2 adrenergic receptor antagonist yohimbine and GRAB-DA mice were injected with the $D_2$ receptor antagonist eticlopride. These antagonists bind to the sensors and prevent them from responding to the presence of NE or DA (*Feng et al., 2019*; *Sun et al., 2020*). Thus, even though yohimbine is known to increase NE levels in the hippocampus (*Abercrombie et al., 1988*), its blockade effect on the GRAB-NE sensor should result in a decrease in fluorescence after administration.

Interestingly, the results for the dopamine and norepinephrine sensors were similar. First, in all groups of animals, there was a small increase in fluorescence immediately after the injection likely produced by a LC response to the highly salient I.P. injection process (data not shown). However, after this increase subsided, fluorescence continued to decrease for the rest of the session in the vehicle and CNO +antagonist groups for both sensors. This continued decrease is likely the result of a reduction in the amount of dopamine and norepinephrine in the HPC (as the chamber becomes less novel and the animals explore less) in addition to any photobleaching that occurred over the session. In contrast, CNO-only injections in mice expressing either sensor produced prolonged increases in fluorescence relative to the vehicle groups beginning approximately 10 min after the injection (*Figure 4C and D*), indicating that stimulation of the LC increases both dopamine and norepinephrine concentrations in the dHPC. The time course of these increases is consistent with the latency of onset of hM3Dq effects on physiology and behavior in other studies (*Alexander et al., 2009*; *Jendryka et al., 2019*). These data confirm that the LC activation can drive the release of both dopamine and norepinephrine into the dHPC. LC activation likely triggers co-release of DA and NE directly from LC terminals in dHPC, but we cannot fully rule out the possibility that the increase in DA is driven indirectly by LC projections to midbrain dopaminergic regions (*Schank et al., 2006*; *Schwarz and Luo, 2015*).

## Dopamine, not norepinephrine, is required for trace fear memory formation

Given that LC activation drives the release of both NE and DA in the dHPC, we sought to determine which of these neurotransmitters contributes to trace fear memory formation. To test this, we trained animals in trace fear conditioning after administering either norepinephrine or dopamine receptor antagonists and tested their memory the next day (*Figure 6A*).

To determine whether norepinephrine is critically involved in trace fear conditioning, we administered a high dose (20 mg/kg) of the β-adrenergic receptor antagonist propranolol 30 min before training. Similar or lower doses of propranolol disrupt memory acquisition and consolidation in a number of tasks, including fear conditioning, spatial recognition memory, inhibitory avoidance, and object recognition (*Díaz-Mataix et al., 2017*; *Przybyslawski et al., 1999*; *Sun et al., 2011*; *Villain et al., 2016*). However, during the memory test the next day, we observed no significant differences in freezing between the propranolol-treated mice and the saline-treated controls (*Figure 6B*), suggesting that norepinephrine acting through β-adrenergic receptors is not required for trace fear memory. This result is consistent with prior work showing that the activation of β-adrenergic receptors is important for the retrieval, but not the acquisition or consolidation of hippocampus-dependent fear conditioning (*Murchison et al., 2004*).

To provide a more complete blockade of the effects of norepinephrine, we also tested whether simultaneous antagonism of both β- and α-adrenergic receptors affects trace fear conditioning. We administered combined injections of propranolol and the α1-adrenergic receptor antagonist prazosin at two different doses (low = 0.5 mg/kg prazosin +5 mg/kg propranolol; high = 1 mg/

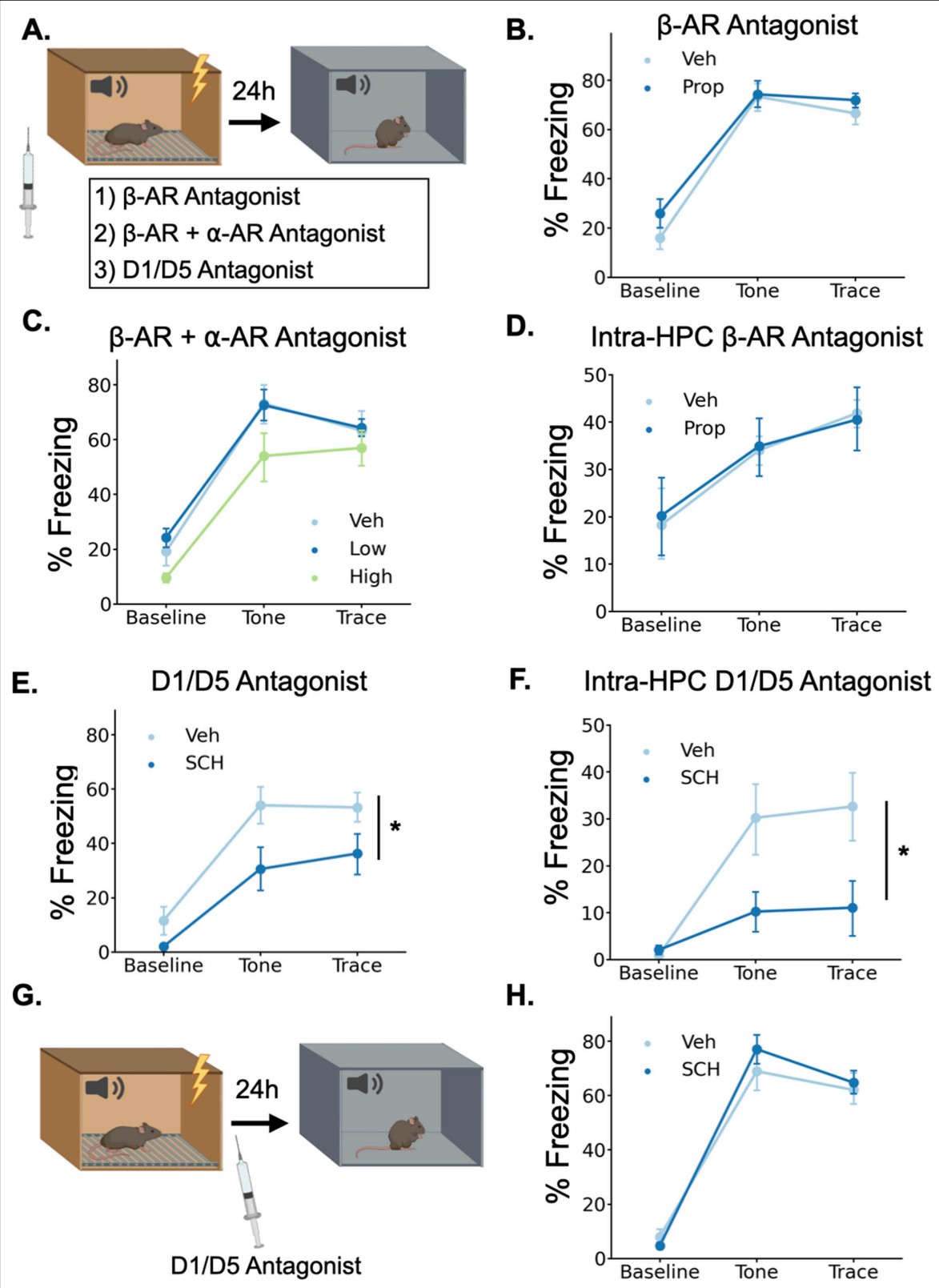

**Figure 6.** Dopamine, not norepinephrine, is required for trace fear memory formation. (**A**) Mice were injected with either propranolol, propranolol and prazosin, or SCH23390 thirty minutes before undergoing trace fear conditioning. The next day, mice were tested for their memory of the tone-shock association in a novel context. (**B**) Freezing behavior during the tone test did not differ between animals that received propranolol injections (Prop, n=8) and vehicle controls (Veh, n=8) (Drug main effect F(1,14) = 1.15, p>0.05; Drug x Epoch interaction F(2, 28)=0.55, p>0.05). (**C**) Freezing behavior during

*Figure 6 continued on next page*

*Figure 6 continued*

the tone test did not differ between animals that received propranolol +prazosin (low (n=6) and high (n=6) doses) and in vehicle controls (n=6)(Drug main effect $F_{(2,15)}$ = 2.26, p>0.05; Drug x epoch interaction $F_{(4, 30)}$=0.59, p>0.05). (**D**) Freezing behavior during the tone test did not differ between animals that received intra-hippocampal propranolol infusions (n=8) prior to training and vehicle controls (n=9) (drug main effect $F_{(1,15)}$ = 0.00, p>0.05; drug x epoch interaction $F_{(2,30)}$ = 0.11, p>0.05). (**E**) Animals that received SCH23390 (SCH, n=8) injections froze significantly less than vehicle controls (Veh, n=8) (Main effect of drug $F_{(1,14)}$ = 6.14, p<0.05). (**F**) Animals that received intra-hippocampal infusions of SCH23390 (n=5) froze significantly less than vehicle controls (n=5) (Treatment x Phase interaction $F_{(2,16)}$ = 4.62, p<0.05). (**G**) Mice were injected with SCH23390 immediately after training and tested the next day. (**H**) There was no significant difference between mice injected with SCH23390 (n=8) after training and vehicle controls (n=8) (Main effect of treatment: $F_{(1, 14)}$=0.175, p>0.05; Treatment X Phase interaction: $F_{(2,28)}$ = 1.65, p>0.05).

kg prazosin +10 mg/kg propranolol) 30 min before training. We found no significant effect of the drug treatments on memory retrieval the next day (*Figure 6C*). Finally, to rule out the possibility that hippocampus-specific blockade of adrenergic signaling is more detrimental to memory formation than systemic blockade, we implanted cannulae in dorsal CA1 and infused propranolol directly into the hippocampus before trace conditioning. We found no significant effect of intra-hippocampal propranolol infusions on long-term memory (*Figure 6D*).

To test whether dopamine is required for trace fear memory formation, we administered the dopamine D1 receptor antagonist SCH23390 (0.1 mg/kg) 30 min before training, as D1/D5 receptors have previously been shown to be critical for other types of hippocampus-dependent memory and plasticity (*Frey et al., 1990*; *Huang and Kandel, 1995*; *O'Carroll et al., 2006*; *Wagatsuma et al., 2018*). During the memory test the next day, SCH-treated mice froze significantly less than the saline controls (*Figure 6E*). To determine whether D1 receptors in the dHPC, specifically, are required for trace fear memory formation, we implanted cannulae bilaterally in dorsal CA1 and infused SCH23390 directly into the hippocampus before trace conditioning. This significantly impaired long term fear memory (*Figure 6F*). Taken together, these data indicate that dHPC dopamine, but not norepinephrine, is critical for trace fear memory formation.

In some memory paradigms, dopamine release in the dHPC has been suggested to mediate the consolidation, rather than the encoding of memories (*Broussard et al., 2016*; *O'Carroll et al., 2006*; *Rossato et al., 2009*; *Takeuchi et al., 2016*). To determine whether the observed memory deficits resulted from impairments in encoding or consolidation, we administered systemic SCH23390 immediately after trace fear conditioning and tested memory the next day (*Figure 6G*). We found no differences in memory performance between SCH23390 and vehicle mice during the memory test (*Figure 6H*), suggesting that the memory impairment observed after pre-training D1 antagonist injections is caused by effects on memory encoding, not consolidation.

## Dopamine release in the dorsal hippocampus during trace fear conditioning

Because we found that LC activation can drive dopamine increases in dHPC and that dopamine release in the dHPC is required for trace fear memory formation, we next examined phasic DA release in the dHPC. To determine whether dHPC DA release increases with stimulus intensity, as LC $Ca^{2+}$ activity does, we infused GRAB-DA3h into dCA1 and measured dopamine transients in response to foot shocks of varying intensities. We found that dHPC dopamine increases in response to foot shocks and that the size of the dopamine transient increases with foot shock amplitude (*Figure 7B and C*), paralleling LC responses.

Next, we measured dHPC dopamine release during trace fear conditioning. Similar to the activity of LC-dHPC axons, dHPC dopamine increased at the tone onset, tone termination, and shock (*Figure 7D*). We also sought to determine whether the dopamine response to the tone or shock changes across learning trials. If dopamine encodes a prediction error, as observed in the VTA then we would expect to see increased responding to the tone and decreased responding to the shock across trials. While dopamine release to the shock did decrease across the training session (*Figure 7E*), there was no change in the size of the response to the tone (*Figure 7F*). To further rule out prediction error signaling in the dHPC dopamine signal, mice underwent a trace conditioning session in which half of the trials were unsignaled (i.e. there was no CS prior to shock) and half were signaled (*Figure 7G*). If

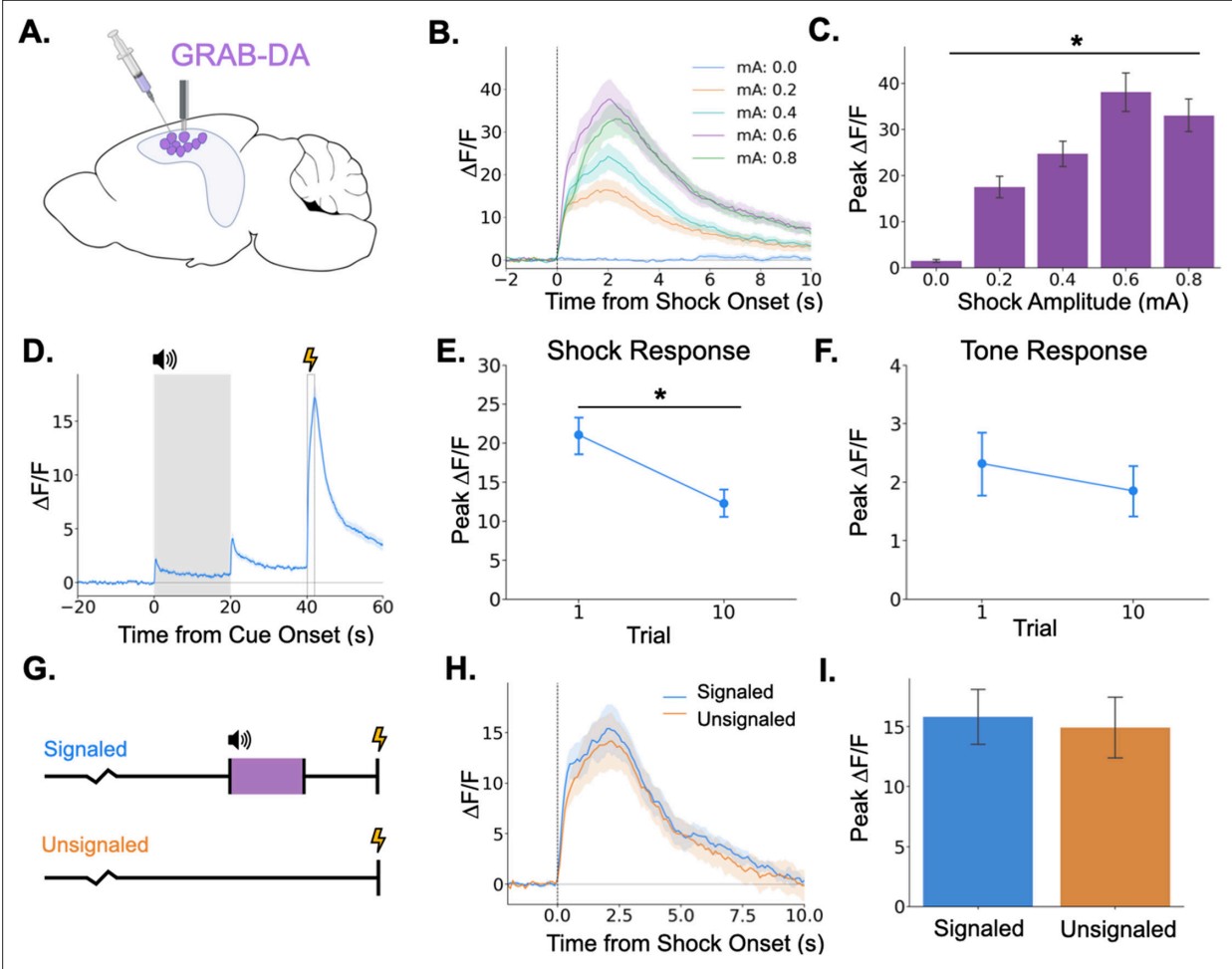

**Figure 7.** Dopamine release in the hippocampus is consistent with release from locus coeruleus axons. (**A**) GRAB-DA3h was infused into the dorsal hippocampus and an optical fiber was implanted in CA1. (**B**) Fiber photometry traces of GRAB-DA responses to shock onset at varying mA levels. Dashed line indicates shock onset. (**C**) Peak ΔF/F during shock onset differed across shock amplitudes (n=10) (F(4,36) = 42.4, p<0.001). (**D**) GRAB-DA photometry traces during trace fear conditioning (n=10). Dopamine increased significantly at tone onset (t(9) = 6.35, p<0.001), tone termination (t(9) = 5.89, p<0.001), and shock (t(9) = 13.0, p<0.001). (**E**) The shock response decreased significantly between trials 1 and 10 (t(9) = 2.42, p<0.05). (**F**) The tone response did not change across trials (t(9) = 0.65, p>0.05). (**G**) Signaled trials were standard trace fear conditioning trials, a twenty-second tone was followed by foot shock after a twenty-second trace interval. Unsignaled trials were the same length, but no tone CS was present before shock. (**H**) Average photometry traces for the shock response in signaled vs unsignaled trials (n=10 animals; 5 trials signaled, 5 unsignaled). (**I**) There was no difference in dopamine response on signaled versus unsignaled trials (t(4) = 1.76, p>0.05).

dHPC dopamine signaled prediction error, we would expect larger dopamine responses to the shock on unsignaled trials compared to signaled trials. However, we found no difference in dopamine transient amplitude between trial types (*Figure 7H, I*).

## Discussion

The locus coeruleus and its interaction with the hippocampus are known to be critical for hippocampus-dependent memory formation (*Compton et al., 1995*; *Kempadoo et al., 2016*; *Lemon et al., 2009*; *Takeuchi et al., 2016*; *Wagatsuma et al., 2018*). However, due to the predominant use of spatial memory tasks in which the learned information is experienced diffusely in time, the precise temporal dynamics of learning-related LC and LC-HPC activity remain poorly understood. In these experiments, we used trace fear conditioning to examine the contribution of precisely timed phasic LC and LC-HPC activity to long-term memory formation.

In the first experiment, we demonstrated that the LC exhibits responses to both neutral and aversive stimuli. This data is consistent with previous reports (*Aston-Jones and Bloom, 1981*; *Hirata and Aston-Jones, 1994*; *Rasmussen and Jacobs, 1986*). Given their relatively short duration and the fact that they are elicited specifically by salient sensory stimuli, we refer to these responses as 'phasic responses'. However, because of the comparatively slow dynamics of fluorescent sensors relative to electrophysiology, we cannot rule out the possibility that these responses are somehow different in nature from previously reported phasic LC responses. Thus, some care must be taken in conflating the characteristics and/or function of the relatively short-lasting responses presented here and the extremely fast phasic responses to very brief (µs to ms) sensory stimuli reported previously. Our data also demonstrate that the size of the LC phasic response is modulated by the intensity, or salience, of sensory stimuli. This relationship between salience and LC response magnitude was particularly pronounced for the aversive foot-shock. Across all intensities, LC responses were larger to the aversive stimuli than to neutral auditory stimuli, suggesting that the LC may encode some information about the emotional valence of stimuli in addition to the simple sensory salience – though to determine this conclusively would require equating the salience of stimuli across the somatosensory and auditory modalities while varying the emotional valence, which may not be possible, especially in a rodent model. These data are consistent with previous work suggesting a role for LC in signaling salience (*Foote et al., 1980*; *Grant et al., 1988*; *Vazey et al., 2018*) and extend them by providing a parametric characterization of LC responses to stimuli of varying intensities.

We also found that phasic LC responses change with learning. First, consistent with a role for LC in signaling salience or novelty, LC responses to a neutral auditory stimulus habituate across repeated exposure to a stimulus, as demonstrated previously (*Hervé-Minvielle and Sara, 1995*). Additionally, we demonstrated that the LC response to the same auditory stimulus is larger when the stimulus is associated with an aversive outcome. Previous work has shown that LC neurons can respond more to conditioned than non-conditioned stimuli (*Aston-Jones et al., 1994*; *Bouret and Sara, 2004*; *Uematsu et al., 2017*); however, to our knowledge, this is the first demonstration of increased LC responding to the CS in a trace conditioning task where the CS and US are separated by an extended interval. This may indicate some level of top-down influence over the LC by the hippocampus or prefrontal cortex, as both of these regions are required for learning in this task (*Gilmartin and Helmstetter, 2010*; *Wilmot et al., 2019*).

Our results also confirmed that the direct projections from the LC to the dHPC are phasically activated by all learning-relevant stimuli during trace conditioning. This data provides support for the idea that phasic LC input to the hippocampus is involved in memory formation. We did not observe obvious changes in tonic LC-HPC activity in our data, although these changes may be more subtle and occur over longer time scales that would be difficult to detect using fiber photometry. The presence of the same responses in the LC-HPC axon terminals that were seen in LC cell bodies fits with the canonical view of the LC as a largely homogenous structure whose neurons' axons are highly collateralized. However, it was important to test whether LC-HPC projections respond during trace conditioning directly, as more recent work suggests the LC may be composed of many distinct 'modules' of neurons that project to separate regions and it is not yet known whether these modules exhibit different response properties (*Schwarz and Luo, 2015*; *Uematsu et al., 2017*). Because we only recorded from LC projections to the HPC, our data could be consistent with either a homogenous or modular LC organization. Future work could distinguish between these possibilities by recording LC projections in an array of downstream targets.

We also sought to determine whether LC stimulation results in the release of dopamine and/or norepinephrine into the HPC. Using DREADDs, we were able to show that direct LC stimulation drives increases in both dopamine and norepinephrine in the HPC, consistent with a number of recent studies indicating their co-release from LC terminals in the hippocampus and neocortex (*Devoto and Flore, 2006*; *Kempadoo et al., 2016*; *Smith and Greene, 2012*; *Takeuchi et al., 2016*; *Wagatsuma et al., 2018*). Due to the low number of VTA/SN fibers in dorsal HPC, it is likely that the large increase in dopamine during LC stimulation is due to direct release from LC terminals. However, it is also possible that HPC dopamine is increased indirectly via LC projections to dopaminergic midbrain regions, although there are several problems for this argument (*Guiard et al., 2008*). First, recent studies have shown that activation of LC terminals in the dorsal hippocampus induces the release of dopamine (*Kempadoo et al., 2016*). Second, inactivation of LC but not VTA terminals in the dorsal

hippocampus decreases dopamine release in this region (*Gálvez-Márquez et al., 2022*). Therefore, while it is not possible to completely rule out increased hippocampal dopamine via a LC-midbrain-HPC circuit based on our studies, direct release of dopamine from LC terminals in HPC is more likely.

A wide body of evidence suggests that LC activation enhances memory acquisition (*Kempadoo et al., 2016*), consolidation (*LaLumiere et al., 2003*; *Novitskaya et al., 2016*; *Takeuchi et al., 2016*), and retrieval (*Murchison et al., 2004*; *Sara and Devauges, 1988*). Here, we extended this data by showing that precisely timed phasic activation of the locus coeruleus enhances memory formation. Specifically, our data indicate that phasic LC activation can enhance memory formation when the relevant stimuli are not salient enough to produce significant learning on their own.

There are multiple possibilities as to the mechanism underlying this enhancement. First, both dopamine (*Frey et al., 1990*; *Lisman et al., 2011*) and norepinephrine (*Bliss et al., 1983*; *Hu et al., 2007*; *Stanton and Sarvey, 1985*) enhance synaptic plasticity in the hippocampus and in other structures important for trace fear conditioning, including the prefrontal cortex and amygdala (*Bissière et al., 2003*; *Huang et al., 2000*; *Tully et al., 2007*). Given our data showing that dopamine, and not norepinephrine, is required for trace fear memory formation, the most direct possibility is that the release of dopamine from the LC enhances learning-related plasticity in regions supporting trace fear conditioning. Our finding that systemic β-adrenergic receptor antagonism has no effect on trace fear memory formation was surprising in light of previous work demonstrating memory impairments after infusions of adrenergic antagonists into the hippocampus or amygdala (*Giustino and Maren, 2018* for review). However, our data is consistent with recent findings from several other groups (*Kempadoo et al., 2016*; *Murchison et al., 2004*; *Takeuchi et al., 2016*; *Wagatsuma et al., 2018*). In particular, the work of Thomas et al. used several different methods to demonstrate that β-adrenergic receptors in the dHPC are required for the retrieval of context fear but not learning or memory consolidation (*Murchison et al., 2004*). Based on this previous work and our data showing learning-related activity in LC-HPC projections, we believe that LC release of dopamine into the hippocampus is at least partially responsible for our observed effects. However, future work manipulating this projection directly would be required to make this conclusion.

The LC may also assert its effect on memory formation indirectly via its influence over sensory processing, attention, or valence processing, all of which are affected by phasic LC activity (*Aston-Jones and Cohen, 2005*; *Bouret and Sara, 2004*; *McCall et al., 2015*; *Vazey et al., 2018*). Under this hypothesis, the LC could facilitate memory by enhancing the responses of neural populations involved in processing, attending, or assigning valence to the learning-relevant stimuli. For example, in our optogenetic experiment (*Figure 4*) phasic LC stimulation may have enhanced stimulus detection and allowed the relatively weak stimuli that were used to capture the animals' attention and produce learning when they otherwise would not. Distinguishing between these plasticity and cognitive modulation accounts of LC-driven memory enhancements would be difficult as many of the same cellular mechanisms are likely involved in both processes. Indeed, it is likely that LC enhancements of plasticity and cognitive function both contribute to its effects on memory.

## Materials and methods

### Subjects

Subjects in this study were 2- to 4-month-old male and female F1 hybrids generated by breeding TH-Cre mice maintained on a C57BL/6 J background (Jackson Labs, Cat #008601) with 129S6 mice (Taconic). Mice were maintained on a 12 hr light/12 hr dark cycle with ad libitum access to food and water. All experiments were performed during the light portion (7 a.m-7 p.m.) of the light/dark cycle. Mice were group housed throughout the experiments. All experiments were reviewed and approved by the UC Davis Institutional Animal Care and Use Committee (IACUC, Protocol #21450).

### Surgery

Stereotaxic surgery was performed 2–3 weeks before behavioral experiments began for LC cell body experiments and 8–14 weeks before behavioral experiments for LC projection experiments. Mice were anesthetized with isoflurane (5% induction, 2% maintenance) and placed into a stereotaxic frame (Kopf Instruments). An incision was made in the scalp and the skull was adjusted to place bregma and lambda in the same horizontal plane. Small craniotomies were made above the desired injection site

in each hemisphere. AAV was delivered at a rate of 2 nl/s to the locus coeruleus (AP – 5.5 mm and ML ± 0.9 mm from bregma; DV –3.6 mm from dura) or dorsal CA1 (AP –2.0 mm and ML ± 1.5 mm; DV – 1.25 mm from dura) through a glass pipette using a microsyringe pump (UMP3, World Precision Instruments).

For optogenetic stimulation experiments, the AAV used was AAV9-EF1A-DIO-hChR2(E123T/T159C)-eYFP (300 nl/hemisphere, titer:8.96x10$^{13}$, Addgene). For DREADDs stimulation epxeriments, the AAV used was AAV9-PRSx8-hM3Dq-HA (300 nl, titer: 2.24x10$^{14}$, gift from Gary Aston-Jones). For photometry experiments, the constructs were AAV5-FLEX-hSyn-GCaMP6s (300 nl, gift from Lin Tian), AAV9-hSyn-FLEX-axonGCaMP6s (300 nl, Addgene), AAV9-hSyn-GRAB-NE1h (250 nl, Addgene), AAV9-hSyn-GRAB-DA2h (250 nl, Addgene), or AAV9-hSyn-GRAB-DA3h (300 nL, gift from Yulong Li).

After AAV infusions, an optical fiber (200 μm diameter for optogenetics, Thorlabs; 400 μm diameter for photometry, Doric) was implanted above dorsal CA1 (dCA1) (AP –2.0 mm and ML ± 1.5 mm from bregma; DV –1.0 mm from dura for optogenetics, DV – 1.25 mm from dura for photometry) or locus coeruleus (AP –5.5 mm and ML –0.9 mm from bregma; DV –3.5 mm). The fiber implants were secured to the skull using dental adhesive (C&B Metabond, Parkell) and dental acrylic (Bosworth Company).

## Apparatus

The behavioral apparatus has been described previously (*Wilmot et al., 2019*). Briefly, fear conditioning occurred in a conditioning chamber (30.5 cm x 24.1 cm x 21.0 cm) within a sound-attenuating box (Med Associates). The chamber consisted of a front-mounted scanning charge-coupled device video camera, stainless steel grid floor, a stainless steel drop pan, and overhead LED lighting capable of providing broad spectrum and infrared light. For context A, the conditioning chamber was lit with both broad spectrum and infrared light and scented with 95% ethanol. For context B, a smooth white plastic insert was placed over the grid floor and a curved white wall was inserted into the chamber. Additionally, the room lights were changed to red light, only infrared lighting was present in the conditioning chamber, and the chamber was cleaned and scented with disinfectant wipes (PDI Sani-Cloth Plus). In both contexts, background noise (65 dB) was generated with a fan in the chamber and HEPA filter in the room.

## Behavioral Procedures

### Trace fear conditioning

All behavioral testing occurred during the light portion of the light/dark cycle. Mice were habituated to handling and optical fiber connection for 3–5 min/day for 5 days before the beginning of behavior. Next, the mice underwent trace fear conditioning in context A. During training, mice were allowed to explore the conditioning chamber for 3 min before conditioning trials began. For optogenetics and pharmacology experiments, the animals then receive 3 trace conditioning trials. For photometry experiments, animals received 10 conditioning trials. Each trial consisted of a 20 s pure tone (85 dB, 3000 Hz) and a 2 s shock (0.3mA for optogenetics and pharmacology, 0.2mA for photometry) separated by a 20 s stimulus-free trace interval. The intertrial interval (ITI) was 120 or 180 s. Mice were removed from the chamber 180 s after the last trial. Twenty-four hours later, the mice were placed in context B for a tone test consisting of a 2 or 3 min baseline period followed by 6 20 s tone presentations separated by a 180 s ITI for optogenetics and pharmacology experiments and 10 20 s tone presentations separated by a 140 s ITI for photometry experiments. In photometry extinction experiments, mice underwent an extinction test in which they received 20 tone presentations separated by 90 second ITI in context B. Freezing behavior was used to index fear and measured automatically using VideoFreeze software (Med Associates).

### DREADDS stimulation behavior

For DREADDS experiments, mice (GRAB-NE, n=5; GRAB-DA, n=6) were placed in the same apparatus used for trace fear conditioning experiments. Baseline fluorescence was acquired for 10 min, after which mice were briefly removed from the chamber to receive an injection of either CNO or vehicle. After injection, mice were immediately placed back inside the chamber and remained there for 50 more minutes. This procedure was repeated the next day, but animals that received CNO on the first day received saline on the second day and vice versa.

### Shock response curve

For the LC shock response curve experiment, mice (n=4) were placed in the conditioning chamber and allowed to explore freely for 180 s. Then 1 s shocks were presented in three blocks of seven trials. Each block consisted of seven trials of different shock intensities (0mA, 0.05ma, 0.1mA, 0.2mA, 0.4mA, 0.6mA, and 0.8mA) in pseudorandom order, such that the shocks did not occur in ascending or descending order or follow the same pattern in each block, separated by a 30 s ITI. Trial blocks were separated by 180 s and mice were removed from the chamber 180 s after the last trial. For the GRAB-DA shock response curve experiment, mice (n=10) underwent 10 trials with two exposures each to 0mA, 0.2mA, 0.4mA, 0.6mA, and 0.8mA with a 120 s ITI.

### Tone response curve

Mice (n=4) were placed in the conditioning chamber and allowed to explore freely for 180 s. Then 25 twenty second tones of varying intensity (55 dB, 65 dB, 75 dB, 85 dB, 95 dB) were presented in pseudorandom order with a 60 s ITI between tones. Mice were removed from the chamber 60 s after the last tone.

### Signaled vs unsignaled fear conditioning

*Animals were trained in TFC as described above. Then, they underwent another behavioral session in which* the procedure was the same as the trace fear conditioning procedure, except that a pseudo-random half of the shocks were unsignaled in that they were not proceeded by a tone CS. These trials would consist of just a 160 s ITI followed by a foot shock. (n=5).

## Optogenetics

Blue light (465 nm, 10 mW measured at fiber tip) was delivered (20 Hz, 5ms pulse width) to LC in 2 s epochs during the training session. Light onset was simultaneous with onset of the tone, termination of the tone, and onset of the shock. No light was delivered during the test. (n=7/group).

## Fiber photometry

The photometry system (Doric) consists of a fluorescence mini-cube that transmits light from a 465 nm LED sinusoidally modulated at ~209 Hz that passed through a 465 nm bandpass filter, and a 405 nm LED modulated at ~308 Hz that passed through a 405 nm bandpass filter. The fluorescence from neurons below the fiber tip was transmitted via this same cable back to the mini-cube, where it was passed through an emission filter, amplified, and focused onto a femtowatt photoreceiver (Newport). The signals generated by the two LEDs were demodulated and decimated to 120 Hz for recording to disk.

All preprocessing and analysis was conducted using custom Python scripts. For preprocessing of GCaMP data, a least-squares linear fit was used to predict 465 nm signal from the 405 nm isosbestic signal. Data from trials was excluded if major artifacts (large, simultaneous deflections in both channels) were present. To calculate a ΔF/F, the predicted values were then subtracted from the true 465 nm signal and this value was divided by the predicted value. In short, the ΔF/F was calculated as below:

$$\frac{\Delta F}{F} = \frac{465 - 465_{predicted}}{465_{predicted}} \times 100$$

For event-based analyses, the ΔF/F was normalized to the baseline of the trial (the 20 s preceding delivery of the tone for fear conditioning, 2 s preceding the tone for tone and shock response experiments). Data was analyzed as the peak ΔF/F during the first 2 s of each stimulus period analyzed. For single group experiments, responses were analyzed for significance by comparing the peak ΔF/F in the two seconds before the event to the 2 s after the event.

For GRAB-NE and GRAB-DA DREADDs experiments, ΔF/F calculation did not use the isosbestic channel due to differential bleaching between the isosbestic and signal channels over the long time scales used in those experiments. Instead, ΔF/F was calculated via baseline normalization (i.e. subtracting the mean 465 nm signal during a 5 min baseline period from the 465 nm signal and dividing the resulting value by the standard deviation of the 465 nm during the baseline period).

## Drugs

For DREADDs experiments, animals received 5 mg/kg I.P. injections of clozapine-N-oxide (CNO, Tocris) dissolved in 2% DMSO in sterile 0.9% saline. Vehicle injections were 2% DMSO in sterile 0.9% saline. All other drugs were dissolved in 0.9% sterile saline. SCH23390 (Sigma) was administered at 0.1 mg/kg I.P. (pre-training: SCH n=8, vehicle n=8; post-training: SCH n=8, vehicle n=8) or intra-hippocampally (5 μg/μL, 0.5 μL/hemisphere) (SCH n=5, vehicle n=5) 30 min before behavioral experiments for pre-training experiments or immediately after animals were removed from the conditioning chamber for the consolidation experiment. Propranolol was administered at 20 mg/kg I.P. (propranolol n=8, vehicle n=8) or intra-hippocampally (10 μg/μL, 0.5 μL/hemisphere; propranolol n=8, vehicle n=9) 30 min before the behavioral session. Combined injections of propranolol and prazosin were administered at either 0.5 mg/kg prazosin and 5 mg/kg propranolol (n=6) or 1 mg/kg prazosin and 10 mg/kg propranolol (n=6) 30 min prior to the behavioral session (vehicle n=6). Yohimbine hydrochloride (Sigma) was administered at 2 mg/kg I.P. 10 min after the start of photometry recordings. Eticlopride hydrochloride (Sigma) was administered at 2 mg/kg I.P. 10 min after the start of photometry recordings.

## Immunohistochemistry and image acquisition

The basic immunohistochemistry procedures have been described previously (*Krueger et al., 2020*; *Wilmot et al., 2019*). Briefly, mice were transcardially perfused with 4% PFA. Following 24–48 hr of post-fixation, 40 μm coronal sections were cut. Slices were washed three times in 1 X phosphate buffered saline (PBS) at the beginning of the procedure and after all antibody and counterstaining steps. All antibodies and counterstains were diluted in a blocking solution containing.2% Triton-X and 2% normal donkey serum in 1 X PBS, unless otherwise indicated. First, sections were incubated for 15 min in the blocking solution. Then, slices were incubated for 24 hr at room temperature in primary antibody. Next, slices were placed in secondary antibody for 60 min at room temperature, followed by detection with a fluorophore for 45 min when required. Finally, sections were stained with DAPI (1:10,000 in PBS, Life Technologies) for 10 min, mounted on slides, and coverslipped with Vectashield anti-fade mounting media (Vector Labs). Images were acquired at ×10–20 magnification using a fluorescence slide scanner (BX61VS, Olympus).

Primary antibodies used included anti-GFP chicken primary antibody (1:300, ab13970, Abcam, RRID:AB_300798), and anti-TH rabbit primary antibody (1:5000, AB152, Sigma). Secondary antibodies included biotinylated donkey-anti-rabbit (1:500, Jackson ImmunoResearch, RRID:AB_2340594), biotinylated donkey-anti-chicken (1:500, Jackson ImmunoResearch, RRID:AB_2313596), and donkey-anti-rabbit-Alexa555 (1:500, Fisher, RRID:AB_162543). Detection was performed with Streptavidin-Cy2 (1:500, Jackson ImmunoResearch, RRID:AB_2337246), Streptavidin-Cy3 (1:500, Jackson ImmunoResearch, RRID:AB_2337244), or Streptavidin-Cy5 (1:500, Jackson ImmunoResearch, RRID:AB_2337245).

## Statistical analyses

Sample sizes were based on previous studies using similar methods. In all experiments, animals were randomly assigned to groups prior to experimentation. For analysis of behavioral data from training and tone test sessions, freezing scores in each phase type (baseline, tone, trace) were averaged across trials for each animal. All data were analyzed using repeated-measures ANOVA, paired t-tests, or unpaired t-tests as appropriate. ANOVA was followed by Bonferroni-corrected post hoc comparisons when necessary. A threshold of $p < 0.05$ was used to determine statistical significance. All data are shown as mean ± SEM. Data were analyzed with custom Python scripts and figures were generated using custom Python scripts and BioRender.

# Additional information

## Funding

| Funder | Grant reference number | Author |
|---|---|---|
| National Institute of Neurological Disorders and Stroke | R01NS129217 | Brian Joseph Wiltgen |
| National Institute of Mental Health | R21MH126496 | Brian Joseph Wiltgen |
| National Institute of Mental Health | T32MH112507 | Jacob H Wilmot |

The funders had no role in study design, data collection and interpretation, or the decision to submit the work for publication.

## Author contributions

Jacob H Wilmot, Conceptualization, Data curation, Formal analysis, Funding acquisition, Investigation, Visualization, Writing – original draft, Writing – review and editing; Cassiano RAF Diniz, Ana P Crestani, Jacob Roshgadol, Investigation, Writing – review and editing; Kyle R Puhger, Software, Writing – review and editing; Lin Tian, Resources, Writing – review and editing; Brian Joseph Wiltgen, Conceptualization, Supervision, Funding acquisition, Writing – original draft, Writing – review and editing

## Author ORCIDs

Jacob H Wilmot ⓘ http://orcid.org/0000-0002-5408-732X
Jacob Roshgadol ⓘ http://orcid.org/0000-0003-3541-0715
Lin Tian ⓘ http://orcid.org/0000-0001-7012-6926
Brian Joseph Wiltgen ⓘ http://orcid.org/0000-0002-3811-5183

## Ethics

All experiments were in accordance with the National Institutes of Health guidelines and reviewed and approved by the UC Davis Institutional Animal Care and Use Committee (IACUC, Protocol #21450).

Reviewer #3 (Public review): https://doi.org/10.7554/eLife.91465.3.sa1
Author response https://doi.org/10.7554/eLife.91465.3.sa2

# Additional files

## Supplementary files

• MDAR checklist

## Data availability

Data and code used in this study are available at https://github.com/jhwilmot/Wilmot-et-al.-2023-Data (copy archived at *Wilmot, 2024*). Up-to-date code for fiber photometry processing and analysis are available at https://github.com/kpeez/fiberphotopy (copy archived at *Puhger, 2024*).

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
