## [Editor Report · eLife assessment]

This is an **important** study examining the neural profile of weak and strong fear memories using a variety of imagining and interrogation neural techniques. The data are **convincing** in detailing the neural profile of neutral, aversive and fear conditioned stimuli in the LC and its input to the dorsal hippocampus and support the conclusion that dopaminergic input from the LC is the key instigator of trace fear conditioning in hippocampus. This paper is of interest to behavioural and neuroscience researchers studying learning, memory and neural networks.

---

## [Referee Report · Reviewer #3 (Public review)]

Summary:

The manuscript examines an important question, namely how the brain associates events spaced in time. It uses a variety of neural methods including fiber photometry as well as area-specific and pathway-silencing methods with the exquisite dissociation of norepinephrine and dopamine. The data show that neurons in the locus coeruleus (LC) respond to auditory cue onset, offset, and shock. These responses are stronger if the cue is paired with shock in a trace procedure. Optogenetic stimulation similar to the neural response captured by fiber photometry enhances associative learning. LC terminals in the dorsal hippocampus also showed phasic responses during fear conditioning and drove dopamine and norepinephrine responses. Pharmacological methods revealed that dopamine and not norepinephrine are critical for fear learning.

Strengths:

The examination of the neural signal to different tone intensities, different shock intensities, repeated tone presentation (habituation), and conditioning, offers an unprecedented account of the neural signal to non-associative and associative processes. This kind of deconstruction of the elements of conditioning offers a strong account of how the brain processes the stimuli used and their interaction during learning.

Excellent use of data acquired with fiber photometry in the optogenetic interrogation study.

The use of pharmacology to disentangle dopamine and norepinephrine was excellent.

Comments on revised version:

The authors have thoroughly and thoughtfully addressed my prior concerns.

---

## [Author Response]

The following is the authors’ response to the original reviews.

**Reviewer 1**
One criticism the authors have made of previous studies was that they have not distinguished between 'tonic' and 'phasic' LC activity and could not demonstrate 'time- locked phasic firing'. This has not been achieved in the present report, as an examination of Fig 1 C,D and 2 C,D shows. Previous reports in rats and monkeys, using unit recording in rats and monkeys clearly show that the latency of LC 'phasic' responses to salient or behaviorally relevant stimuli are in the range of tens of milliseconds, with a very short duration, often followed by a long-lasting inhibition. This kind of temporal precision concerning the phasic response cannot be gleaned from the time scale shown in the Figures (assuming the time scale is in seconds). We can discern a long-lasting increase in tonic firing level for the more salient stimuli (Fig 1C) (although the authors state in the discussion that "we did not observe obvious changes in tonic LC-HPC activity). This calcium imaging methodology as used in the present experiments can give us a general idea of the temporal relation of LC response to the stimulus, but apparently does not afford the millisecond resolution necessary to capture a phasic response, at least as the data are presented in the Figures.

While we understand the reviewer’s concern with our use of the terms phasic and tonic, we believe we have represented them as accurately as possible given our data. Unfortunately, the distinction between tonic and phasic activity is somewhat arbitrary, in that there is no strict definition, to our knowledge, of the exact parameters that activity must fall into to be categorized as tonic or phasic. While it is true that phasic LC activity has typically been studied with electrophysiological approaches that afford millisecond resolution and that observed phasic responses are often extremely short, there are numerous differences between those studies and this one. Most prominently, the stimuli used to elicit a phasic response are generally extremely short (often 1ms or less) and therefore generate extremely short phasic responses (Aston-Jones and Bloom, 1981a; Aston-Jones and Cohen, 2005), but this is not to say that phasic responses might not be longer in response to a longer lasting stimulus. Moreover, tonic activity is reported to track with behavioral state on the order of dozens of seconds to minutes and is not reported in response to specific stimuli (Aston-Jones and Bloom, 1981b). The “phasic” responses we report generally decay in less than 5 seconds in our fluorescence signals. Given the slow time course of decay for GcAMP6s (a single action potential can generate a response that lasts 3 or more seconds (Chen et al., 2013)) and the GRAB sensors (GRAB-DA2h τoff = 7.2s (Sun et al., 2020)), the underlying neural responses would have lasted for a significantly shorter period. Therefore, we believe the responses we observed are much more consistent with phasic responses to long-lasting sensory stimuli (20-second tone, 1-2 second shock), than with increases in tonic activity associated with a change in behavioral state. Finally, regardless of whether these responses are exactly the same as previously reported phasic responses, our photometry and optogenetics studies provide insight about a form of LC activity that is fundamentally different than what can be gleaned from much slower dialysis, lesion, and pharmacology studies. Nonetheless, we added the following to the discussion section to clarify the limitations of our interpretation:

“…given their relatively short duration and the fact that they are elicited specifically by salient sensory stimuli, we refer to these responses as “phasic responses.” However, because of the comparatively slow dynamics of fluorescent sensors relative to electrophysiology, we cannot rule out the possibility that these responses are somehow different in nature to previously reported phasic LC responses. Thus, some care must be taken in conflating the characteristics and/or function of the relatively short-lasting responses presented here and the extremely fast phasic responses to very brief (μs to ms) sensory stimuli reported previously.”

Much of the data presented here can be regarded as 'proof of concept' i.e. demonstrating that Photometric imaging of calcium signalling yields similar results concerning LC responses to salient or behaviorally relevant stimuli as has been previously reported using electrophysiological unit recording. The role of dopamine as the principal player in hippocampaldependent learning also corroborates previous reports.

Although some of the data presented in this study could be seen as “proof of concept” or “confirmatory” of previous results, we believe this work extends previous results by showing (1) the importance of hippocampal dopamine to aversive hippocampus-dependent learning and trace fear conditioning specifically, (2) that LC responses are important at the specific times of learning (i.e. CS/US onset/termination), and (3) that dopamine in the hippocampus is likely important for learning in a way that is not tied to prediction error or memory consolidation.

No attempt was made to address the important current question of the modular organisation of Locus Coeruleus, although the authors recognize the importance of this question and propose future experiments using their methodology to record simultaneously in several LC projection sites.

While we do recognize the importance of this modular organization, which is addressed in the discussion as the reviewer mentions, experiments addressing this organization are beyond the scope of the present study. Future work will address the possibility that LC projections to different regions show differential responses during learning.

The phasic-tonic issue has not been resolved by these experiments. Phasic responses of LC single units are short-latency, short-lived (just 3-4 action potentials), and followed by a relatively long refraction period. Multiunit responses will have a more jittery latency and longer-lasting response (but still only tens to hundreds of milliseconds). Your figures clearly show long-lasting increases in tonic firing levels, even though you state the contrary in the discussion. Therefore, I strongly recommend removing the word 'phasic' from the title.

Addressed above.

Yohimbine, the Alpha 2 antagonist, administered systemically, induces a massive increase in the rate of firing of LC cells (through blockade of autoinhibition at the cell body level at terminals). I guess its effect on the receptor 'backbones' overrides the massive release of NE and/or DA, but you might want to mention this; also include the dose of all drug treatments.

Yes, yohimbine’s effect on the GRAB-NE signal is somewhat counter-intuitive given the known effect of yohimbine on norepinephrine levels. However, our result is consistent with previous reports (Feng et al., 2019). We have added the following to the results section to clarify:

“Thus, even though yohimbine is known to increase NE levels in the hippocampus (Abercrombie et al., 1988), its blockade effect on the GRAB-NE sensor should result in a decrease in fluorescence after administration.”

Include time scale units on all figures (I assume it is seconds in Figs 1 &2).

Thank you for pointing out this issue, we have added units on all figures.

Is it possible to have a better quality example of staining? Fig 1 B in particular is very blurry. Is the yellow double staining? Please indicate. Most of the GCaMP seems to be outside the main area of TH staining. Fig 4 B is much nicer--and it looks morphologically, like LC.

Unfortunately, the GcAMP6s staining was very dim in our hands and resulted in relatively blurry images. Yes, in this case, yellow is double staining. Regarding the morphology, the GCaMP image is taken from a sagittal section and the shape of expression is consistent with images of LC in the sagittal plane. However, given the quality of our ChR2 images, we are confident in the specificity of expression in these mice.

**Reviewer 2**
The claim that dopamine release in dHPC is caused by LC neurons is not directly tested. Unfortunately, the most critical experiment for the claims that dopamine release comes from LC during conditioning is not tested. A lack of dopamine signal in dHPC caused by inhibition of LC during TFC would show this. It is indeed an interesting observation that chemoegenetic activation of LC causes dopamine release in the dHPC. However, in the absence of concurrent VTA inhibition or lesion, it remains a possibility that the dopamine release is mediated through indirect actions on other dopamine-expressing neurons. The authors do a good job of arguing against this interpretation in the discussion, and the literature seems appropriate for this. However, the title is still an overstatement of the data presented in this study.

We agree with the reviewer’s comments. As indicated in the discussion, it is possible that hippocampal dopamine is increased indirectly via LC projections to dopaminergic midbrain regions. We believe that our title is consistent with this possibility. When phasic stimulation was delivered to the LC, dopamine levels increased in the hippocampus and trace fear conditioning was enhanced. The observed increase in dopamine could be direct or indirect. As the reviewer notes, we argue for the former in the discussion section. A number of experiments would be needed to show this directly (record dopamine while: inhibiting the LC, inhibiting the VTA, stimulating LC while simultaneously inhibiting the VTA etc.) and we are planning to do these in the future.

The primary alternative interpretations of the phasic activation experiment are whether only stimulation to the cue events (both on and off), or whether only stimulation to the shock. Thus this experiment would benefit from additional data showing either a no shock control, to show that enhanced activity of the LC to the tone is not inherently aversive, or manipulations to the tone but not to the shock.

Future work will explore whether the contribution of LC to learning is primarily due to its activation during the CS or the US. However, this is beyond the scope of this manuscript.

Specificity of the GRAB-NE and GRAB-DA sensors should be either justified through additional experiments testing the alternative antagonist (i.e. GRAB-NE CNO+eticloprode / GRAB-DA CNO+yohimbine) or additional citations that have tested this already. It is critical for the claims of the paper to show that these sensors are specific to dopamine or norepinephrine.Although sensitivity is a potential concern, these sensors have been thoroughly vetted and used by many groups since their generation. In particular, the creators of these sensors provided extensive data showing their specificity. The GRAB-DA sensor is ~10 fold more sensitive to DA than to NE (Sun et al., 2020, cited 239 times) and the GRAB-NE sensor is ~37 fold more sensitive to NE than to DA (Feng et al., 2019, cited 371 times).The role of dopamine in prediction error was tested through a series of conditions whereby the shock was presented either signaled (i.e. predicted), or not. However, another way that prediction error is signaled is through the absence of an expected outcome. Admittedly it might not be possible to observe a decrease in dopamine signaling with this methodology.

Although this is a strong point, given that the study is not primarily focused on error prediction and the low likelihood of observing the typically small decrease in signaling during expected outcome omission, we feel that additional error prediction studies are beyond the scope of this manuscript. However, further experiments as suggested by the reviewer could prove interesting in future studies.

The difference between Fig. 6E and 6H needs to be clarified. What is shown in Fig. 6E is that the response to the shock decreases through experience (i.e. by the 10th trial). However in Fig 6H, there is no difference between signaled and signaled shock, but this is during conditioning, and not after learning (based on my understanding of the methods, line 482).

We are not sure we fully understand what point of clarification the reviewer is asking for. However, we have clarified in the methods that the signaled vs unsignaled shock experiment took place in animals that had already been trained on TFC. Thus, all of the trials took place after the animals had learned the tone-shock association. Therefore, although the drop in shock-response could be taken as an indicator of a prediction-error like signal, all the other data points to this not being the case (no change in tone response over training, no difference in signaled vs. unsignaled responses after training).

Unless I missed it, at no point in the manuscript is the number of subjects described. Please add the n per experiment within each section describing each experiment in the methods (Behavioral procedures). Some more details in the photometry statistical analysis would be helpful. For example, what is the n per group for every data set that is presented? How many trials per analysis?

We thank the reviewer for pointing this out. Animal numbers have been added in the methods section in the Behavioral Procedures, Optogenetics, and Drugs sub-sections and in the figure legends. Trial numbers are included in these sections and all trials were used for analysis.

What is the difference in experimental procedure between Fig. 2D and Fig. 3B? It seems that they are the same, and yet the LC response to the conditioned CS is not.

Fig. 3B is simply the Day 1 data from Fig 2D presented at a different scale because the shock response is included in Fig. 3B which necessitates a larger scale on both axes. Close inspection of the figures will show that the shapes of these two curves and the error around them is the same, but the different scaling obfuscates this slightly.

Typo in the legend of Figure 2 - D should be E.

Thank you, we have corrected this.

• Anatomical localization of the virus injections, and more importantly the fiber placements, is not shown. Including this information helps with replication and understanding where exactly the observations were made in dHPC to contrast with prior studies.

Representative examples are included in the manuscript in figure 1B, 3F, 4B, and 5B.

**Reviewer 3**
While the optogenetic study was lovely, a control using the same stimulation but delivered at different time points would have been a good addition to show how critical the neural signal at tone onset, tone offset, and shock is.

We agree that it would be interesting in future studies to delineate the specific times when LC stimulation produces a learning enhancement. It could be that LC activity is most important during one specific time period (eg. just during shock) or that all three periods of activation are required. It would also be useful to know whether stimulation at other times during learning can produce an enhancement given the potentially long-lasting effects of dopamine on HPC plasticity and learning.

Justification for the focus on D1 receptors was lacking.

We chose to focus on D1 receptors because previous studies have shown that these receptors are critical for memory formation or consolidation in the hippocampus. We have added a sentence justifying this in the results section.

“To test whether dopamine is required for trace fear memory formation, we administered the dopamine D1 receptor antagonist SCH23390 (0.1mg/kg) 30 minutes before training, as D1/D5 receptors have previously been shown to be critical for other types of hippocampus dependent memory and plasticity (Frey et al., 1990; Huang and Kandel, 1995; O’Carroll et al., 2006; Wagatsuma et al., 2018).”

The manuscript provides convincing evidence that the neural signal is not an error- correcting one by including a predicted (by a tone) and unpredicted shock. One possibility is that perhaps the unpredicted shock could be predicted by the context. Some clarification on the behavioural procedures would help understand if indeed the unsignaled shock could be predicted by the context or not.

Mice always exhibit freezing in the training environment, so the context is definitely a predictor of shock. However, the tone is a much better predictor because it is always followed by shock while the mice spend a large amount of time in the context without being shocked. This is demonstrated by the fact that the same procedure used in the current experiments consistently produces more tone fear than context fear (Wilmot et al., 2019). While we did not do long-term memory tests here, we assume the same dissociation occurred as it has been observed very consistently across studies (Chowdhury et al., 2005; Kitamura et al., 2014; Wilmot et al., 2019). Nonetheless, it is possible that a difference between signaled and unsignaled groups was obscured by the context. We should note however, that differences between dopaminergic responses to cued and uncued rewards and aversive outcomes has been observed and these animals were also trained in the same context (Eshel et al., 2016; Matsumoto and Hikosaka, 2009; Pan et al., 2005; Schultz, 1998). Therefore, we believe this experiment does differentiate the observed dopamine response in the hippocampus from previously reported VTA dopamine prediction error signaling.

Figure 2 - tone termination in Tone only group - no change? Stats?

Thank you for pointing out this omission. We have added the stats to the figure legend. Although the response to tone termination decreased numerically, it did not change significantly across days. This is one point we may seek to clarify in future studies, as the difference between tone onset and termination responses is unexpected. Given the relatively small responses, it’s possible future studies with stronger signal (eg. GcAMP8) may find differences in the tone termination response across training days. This is one of the reasons we focused primarily on the responses to tone onset and shock in the rest of the manuscript.

Fig 4 data - stimulation at time incongruent with the signal as a control for the timing of stim.

This is addressed above.

Fig 5 - GRAB-NE - yohimbine seems to suppress the signal below the vehicle. Not the case for GRAB-DA. Is this sig? post-hoc stats?

Yes, this does appear to be the case for GRAB-NE, and would not be entirely surprising given that there is likely a baseline level of NE (and dopamine) in the hippocampus that produces some degree of baseline fluorescence in the vehicle group. This signal could be reduced/abolished by blocking the sensor and preventing this baseline level of NE from binding and producing fluorescence. This may not be the same for the GRAB-DA for a variety of reasons – different sensor binding affinities, different baseline neurotransmitter levels, potentially non-equivalent drug doses, etc. Because of the large number of pairwise comparisons in this data (18), we did not make post-hoc pairwise comparisons.

Shock response curve - lines 466-474 - some explanation of what the pseudorandom order of shock presentation means.

We have added the following explanation to this section:

“…pseudorandom order, such that the shocks did not occur in ascending or descending order or follow the same pattern in each block,…”

Line 126 - the extinction came out of the blue, it needs some introduction such as a statement that the animals were exposed to extinction training following conditioning.

We have added the following earlier in that same paragraph:

“On the second and third days, mice underwent extinction trials in which no shocks were administered.”

References in Response

Abercrombie ED, Keller RW, Zigmond MJ. 1988. Characterization of hippocampal norepinephrine release as measured by microdialysis perfusion: Pharmacological and behavioral studies. Neuroscience 27:897–904. doi:10.1016/0306-4522(88)90192-3

Aston-Jones G, Bloom FE. 1981a. Nonrepinephrine-containing locus coeruleus neurons in behaving rats exhibit pronounced responses to non-noxious environmental stimuli. Journal of Neuroscience 1:887–900. doi:10.1523/JNEUROSCI.01-08-00887.1981

Aston-Jones G, Bloom FE. 1981b. Activity of norepinephrine-containing locus coeruleus neurons in behaving rats anticipates fluctuations in the sleep-waking cycle. J Neurosci 1:876–886. doi:10.1523/JNEUROSCI.01-08-00876.1981

Aston-Jones G, Cohen JD. 2005. AN INTEGRATIVE THEORY OF LOCUS COERULEUSNOREPINEPHRINE FUNCTION: Adaptive Gain and Optimal Performance. Annual Review of Neuroscience 28:403–450. doi:10.1146/annurev.neuro.28.061604.135709

Chen T-W, Wardill TJ, Sun Y, Pulver SR, Renninger SL, Baohan A, Schreiter ER, Kerr RA, Orger MB, Jayaraman V, Looger LL, Svoboda K, Kim DS. 2013. Ultrasensitive fluorescent proteins for imaging neuronal activity. Nature 499:295–300.doi:10.1038/nature12354

Chowdhury N, Quinn JJ, Fanselow MS. 2005. Dorsal hippocampus involvement in trace fear conditioning with long, but not short, trace intervals in mice. Behavioral Neuroscience 119:1396–1402. doi:http://dx.doi.org/10.1037/0735-7044.119.5.1396

Eshel N, Tian J, Bukwich M, Uchida N. 2016. Dopamine neurons share common response function for reward prediction error. Nat Neurosci 19:479–486. doi:10.1038/nn.4239

Feng J, Zhang C, Lischinsky JE, Jing M, Zhou J, Wang H, Zhang Y, Dong A, Wu Z, Wu H, Chen W, Zhang P, Zou J, Hires SA, Zhu JJ, Cui G, Lin D, Du J, Li Y. 2019. A Genetically Encoded Fluorescent Sensor for Rapid and Specific In Vivo Detection of Norepinephrine. Neuron 102:745-761.e8. doi:10.1016/j.neuron.2019.02.037

Frey U, Schroeder H, Matthies H. 1990. Dopaminergic antagonists prevent long-term maintenance of posttetanic LTP in the CA1 region of rat hippocampal slices. Brain Research 522:69–75. doi:10.1016/0006-8993(90)91578-5

Huang YY, Kandel ER. 1995. D1/D5 receptor agonists induce a protein synthesis-dependent late potentiation in the CA1 region of the hippocampus. Proceedings of the National Academy of Sciences 92:2446–2450. doi:10.1073/pnas.92.7.2446

Kitamura T, Pignatelli M, Suh J, Kohara K, Yoshiki A, Abe K, Tonegawa S. 2014. Island Cells Control Temporal Association Memory. Science 343:896–901. doi:10.1126/science.1244634

Matsumoto M, Hikosaka O. 2009. Two types of dopamine neuron distinctly convey positive and negative motivational signals. Nature 459:837–841. doi:10.1038/nature08028

O’Carroll CM, Martin SJ, Sandin J, Frenguelli BG, Morris RGM. 2006. Dopaminergic modulation of the persistence of one-trial hippocampus-dependent memory. Learning & memory 13:760–769.

Pan W-X, Schmidt R, Wickens JR, Hyland BI. 2005. Dopamine Cells Respond to Predicted Events during Classical Conditioning: Evidence for Eligibility Traces in the Reward-Learning Network. J Neurosci 25:6235–6242. doi:10.1523/JNEUROSCI.1478-05.2005

Schultz W. 1998. Predictive Reward Signal of Dopamine Neurons. Journal of Neurophysiology 80:1–27. doi:10.1152/jn.1998.80.1.1

Sun F, Zhou J, Dai B, Qian T, Zeng J, Li X, Zhuo Y, Zhang Y, Wang Y, Qian C, Tan K, Feng J, Dong H, Lin D, Cui G, Li Y. 2020. Next-generation GRAB sensors for monitoring dopaminergic activity in vivo. Nat Methods 17:1156–1166. doi:10.1038/s41592-02000981-9

Wagatsuma A, Okuyama T, Sun C, Smith LM, Abe K, Tonegawa S. 2018. Locus coeruleus input to hippocampal CA3 drives single-trial learning of a novel context. Proceedings of the National Academy of Sciences 115:E310–E316. doi:10.1073/pnas.1714082115

Wilmot JH, Puhger K, Wiltgen BJ. 2019. Acute Disruption of the Dorsal Hippocampus Impairs the Encoding and Retrieval of Trace Fear Memories. Frontiers in Behavioral Neuroscience 13. doi:10.3389/fnbeh.2019.00116